# Effects of Social Networks on Job Performance of Individuals among the Hypertension Management Teams in Rural China

**DOI:** 10.3390/healthcare11152218

**Published:** 2023-08-07

**Authors:** Qingyun Xia, Yanyun Xu, Xiang Liu, Yingzi Liu, Jian Wu, Meng Zhang

**Affiliations:** 1School of Public Health, Hangzhou Normal University, Hangzhou 311121, China; xiaqy@gs.zzu.edu.cn (Q.X.);; 2College of Public Health, Zhengzhou University, Zhengzhou 450001, China

**Keywords:** job performance, social network, hypertension management, centrality, structural holes

## Abstract

Background: Limited studies have explored the relationship among cross-organizational and multidisciplinary medical staff. Aim: The present study conducted an in-depth examination and validation of the influence of complex cross-organization and multidisciplinary social networks on the job performance of team members. Method: Multi-level hierarchical regression analysis was used to assess the impact of the centrality and the characteristics of structural holes in social networks (i.e., advice network, information network, friendship network, and trust network) on job performance. Results: The in-closeness centrality of the advice network (β = 0.176, *p* < 0.05) and the betweenness centrality of the trust network (β = 0.126, *p* < 0.05) had positive effects on task performance. The in-closeness centrality of the advice network (β = 0.226, *p* < 0.05; β = 0.213, *p* < 0.05) and the CI (1 − constraint index) of the friendship network (β = 0.130, *p* < 0.05; β = 0.132, *p* < 0.05) had positive effects on contextual performance and overall job performance. Meanwhile, the out-closeness centrality of the information network (β = −0.368, *p* < 0.01; β = −0.334, *p* < 0.05) had a negative effect on contextual performance and overall job performance. Conclusions: This study investigates the relationship between healthcare professionals’ job performance and their social networks, taking into account the perspectives of cross-organizational and multidisciplinary teams. The study contributes to the effort of breaking down barriers between different disciplines and organizations, and ultimately, improving the quality of healthcare delivery.

## 1. Introduction

Integrated healthcare is a key part of the healthcare reform to solve the current problems in the medical field and ultimately improve the health of the population [1]. Many countries have taken various measures to promote the coordination and integration of medical institutions, such as structural, functional, and clinical integration. The integration process involves the medical institution establishing a division of labor and cooperation mechanism to achieve collaborative patient services across specialties, institutions, and support systems. Cross-organizational and multidisciplinary cooperation in the management of non-communicable diseases (NCDs) is an important component of integrated healthcare. It requires multiple institutions and departments to engage in medical and health activities based on common goals and cooperative rules, informed by multi-subject and multi-level coordination [2,3].

In order to better promote the realization of the goal, medical institutions establish corresponding functional teams through mutual cooperation and gradually form a cross-organizational and multidisciplinary communication and cooperation network in the process. Based on the Structure–Conduct–Performance (SCP) analysis paradigm proposed by Bain [4], the logical chain of “structure determines behavior, and behavior determines performance” is established. Specifically, it can be considered that the social network built by individuals within and between organizations determines their various behaviors within and between organizations, and these behaviors ultimately affect individual performance. Several studies have shown that the relationship of a team impacts the performance of medical staff [5,6]. Information sharing, communication coordination, and shared goals among team members can contribute to making high-quality decisions, improving medical interventions, facilitating the implementation of NCD prevention and control measures, and improving primary care outcomes [7].

Social network analysis (SNA) is the process of quantifying and assessing the value of social relationships by studying network relationships using quantitative analysis. The social network suggests that individual behavior depends on the pattern or structure of relationships in which they are embedded, and individual job performance is related to their social network position. Burns et al. used social network research methods to explore the impact of relationship coordination and integration on patient care, emphasized the importance of relationship coordination and integration, and constructed a model to understand the networks of relational coordination within healthcare organizations, which provide a reference for future research [8]. Both social network relationships and structures are strongly related to job performance output in educational and industrial organizations [9,10]. A member’s position in the network has an impact on job performance [11]. Furthermore, members central to the network have higher independence and potential influence on other members [12], thus influencing their task performance (TP) [13], work commitment [14], and interpersonal facilitation [15]. Networks with existing “structural holes” can cause members to occupy more “bridges”, thereby increasing control over resources [16] and resulting in comparatively improved job performance [17]. A central position in a social network can improve direct individual access to work-related information, resources, and psychological support [18]. Furthermore, it can create a more profound impact while improving performance [19], which is most conducive to task completion.

In the medical field, structural integration and functional integration have been extensively studied, while research on process and interpersonal integration needs to be deepened [20]. At the same time, the research on integration effect is scarce. Studying the existing problems of cross-organizational and multidisciplinary cooperation is essential to improve performance levels. In addition, hypertension is a major public health concern globally with an increasingly rapid prevalence in low- and middle-income countries [21]. At present, hypertension management in rural China needs to be addressed through cross-organizational and multidisciplinary collaboration. However, few studies have explored the current situation and effectiveness of communication and cooperation between NCD management teams, especially hypertension management teams. Previous studies have underscored the impact of social network characteristics on job performance, which provides feasibility for the present research. Additionally, Fu [22] demonstrated the theoretical scientific validity of studying the effect of social networks on the job performance of hypertension management teams at the county, township, and village levels. Therefore, starting from NCDs, this study uses complex network theory to explore the impact of cross-organizational and multidisciplinary cooperation social networks on the job performance of medical staff to provide evidence support for the integration effect and reference for future research.

## 2. Methods

### 2.1. Study Design and Participants

A cross-sectional survey was conducted from April 2021 to June 2021. The county medical alliance developed the integrated three-level county territory medical service system. This system underscored the exploration of the integrated county–township–village management of a county territory with “county-level hospitals as leaders, township health centers as hubs, and village health clinics as bases”, as well as effective connection with the township–village integration [23]. In this study, the World Bank China Health Reform Program-for-Results provinces—Fujian and Anhui—were randomly selected as medical alliance pilot counties (Suixi County in Anhui Province and Youxi County in Fujian Province). A quantitative analysis was conducted on the social networks of the hypertension management teams in Suixi County Hospital Medical Alliance, Suixi County Traditional Chinese Medicine Hospital Medical Alliance, and Youxi County General Hospital Medical Alliance. Ultimately, we conducted a questionnaire survey based on the hypertension management team member list provided by the personnel department of the county medical alliance.

Members of the hypertension management team of the county medical alliance, such as physicians from cardiology departments of county-level hospitals, general practitioners, nurses, and public health physicians from township hospitals or community health service centers, and village doctors from village clinics, were selected as the study participants. Further, we completed drawing and numbering all the codes of the study participants in the investigation. A total of 382 questionnaires were distributed, of which 338 were completed and returned, resulting in a response rate of 88.48%. Ethical approval for this study was obtained from the Ethics Committee of Hangzhou Normal University.

### 2.2. Measures

Prior studies have shown that age [24], marital status [25], health condition [26], level of health services institutions served [27], and years of service [28] all impact job performance. Thus, these demographic variables were considered control variables associated with individual job performance.

#### 2.2.1. Job Performance

Job performance refers to the sum of behaviors and results related to organizational goals within a certain period. TP refers to employee behaviors that are directly related to the organization and technology, reflecting the maintenance and service work that realizes technical needs. CP (contextual performance) refers to voluntary employee behaviors that can facilitate the organization and its members to achieve organizational goals. According to the job performance scale designed by Han [29], two factors—TP (four entries) and CP (five entries)—were used in this study to measure the job performance of hypertension management team members. Overall job performance is the sum of TP and CP scores. Each option was scored using a five-point Likert scale (1 = “strongly disagree”, 2 = “disagree”, 3 = “neither agree nor disagree”, 4 = “agree”, or 5 = “strongly agree”). The score is calculated by the mean of each item. The higher the score, the higher the job performance.

#### 2.2.2. Social Networks

The social networks in this study were classified into four types based on the social network classification methods proposed by Krackhardt and Jiade Luo: the advice network, information network, friendship network, and trust network [30]. The advice network refers to the network of relationships between the adviser and the advisee regarding challenges in hypertension management. The information network represents the degree of formal and informal information transfer among hypertension management team members. The friendship network represents the degree of emotional communication and association among team members. Finally, the trust network represents the degree of trust and dependence among team members. The encoding information contained in the rows and columns represents hypertension management team members, and the content of the cells represents the contacts between two parties. A “1” indicates a connection between two hypertension management team members, while a “0” indicates no connection between two team members.

The social network characteristics of the advice, information, friendship, and trust networks of the hypertension management team members in the county medical alliances were taken as the independent variables. The whole network was examined using two variables: density and average distance. The variables for the ego network were centrality indicators (degree centrality, betweenness centrality, and closeness centrality) and a structural holes indicator (CI).

“*density*” reflects the closeness of the connection between the network members. The greater the density, the closer the connection between the network members and the higher their degree of interaction with each other.
(1)density=L/[N(N−1)]
where *L* denotes the actual number of relationships contained in the network, and *N* (*N* − 1) denotes the theoretical maximum number of relationships within the network.

“Average distance” measures the scale and cohesiveness of the networks of the hypertension management teams at the county–township–village levels. The smaller the average distance, the larger the cohesiveness index based on “distance”, indicating the stronger cohesiveness of the network.

“Degree centrality” measures the number of points directly adjacent to the member at the node, where the higher the degree centrality, the closer the member is to the center of the network. Since absolute degree centrality is limited for analyzing the networks of different scales, the relative degree centrality to measure the degree centrality of different scales was used instead.
(2)$$C_{RD}^{\prime }\left(x\right)=\frac{x^{\prime }\mathrm{in}\text{-}\mathrm{degree}+x^{\prime }\mathrm{out}\text{-}\mathrm{degree}}{2(n-1)}$$
where “*n*” denotes the total number of nodes in the network.

The variable “betweenness centrality” quantifies the number of times that a node acts as a bridge for the minimal distance between two other nodes and represents the extent of control of resources in the network among medical staff. The higher the node’s betweenness centrality, the more the node occupies the critical position for resources and information flow. In this study, the relative betweenness centrality was calculated using the formula below.
(3)$$C_{RBi}=\frac{C_{ABi}}{n^{2}-3n+2}$$
where *C_ABi_* indicates the absolute betweenness centrality, and “*n*” denotes the total number of nodes in the network.

The variable “closeness centrality” refers to the sum of the minimal distances from a node to other nodes. In a directed network, the higher the in-closeness centrality of the node, the easier it is for other nodes to get to it; the higher the out-closeness centrality of the node, the easier it is for it to get to other nodes. Thus, the in-closeness centrality reflects the integration power of the members within the network, and the out-closeness centrality reflects the radiation power of the members within the network [31].
(4)$$C_{RPi}=\frac{n-1}{\sum_{j=1}^{n}d_{ij}}$$
where *d_ij_* indicates direct relations between node *i* and node *j* (i.e., the number of direct relations), and *n* indicates the total number of nodes in the network.

The variable *CI* is the difference of 1 minus the “network constraint index” (i.e., *C_ij_*), measuring the abundance of the structural holes in the network [32]. *C_ij_* is used to measure the scarcity index of structural holes owned by the team on the basis of structural hole theory. Larger levels of *CI* result in fewer network constraints and increased structural holes in the network [16,33].
(5)$$CI_{i}=1-C_{i}=1-\sum_{j}C_{ij}=1-\sum_{j}{\left(p_{ij}+\sum_{q}p_{iq}p_{qj}\right)}^{2}$$
where node *q* is the common adjacency of *i* and *j*. *P_ij_* represents the weight proportion of *j* in all adjacent points of *i*.

### 2.3. Statistical Analysis

SPSS 23, AMOS 24.0 (IBM, Armonk, NY, USA), and UCINET 6.735 (Analytic Technologies, Lexington, KY, USA; 46171595-13776798) were used to conduct the analysis and hypothesis testing of the collated data, which mainly comprised the following statistical methods. First, a goodness-of-fit test was used to determine the consistency between the observed data and theoretical data in the job performance scale. The results are shown in Table 1. All the fit indices reached a good level, and the overall fit of the model was good (see Table A1). Second, confirmatory factor analysis and structural equation modeling were used to analyze the reliability and validity of the responses to the job performance scale by hypertension management team members in the county medical alliances (see Table A2 and Table A3). Third, descriptive statistics methods were used to describe the socio-demographic characteristics of the hypertension management team members at the county–township–village levels.

Then, univariate analysis was used to determine the difference between TP and CP among medical personnel with different demographic characteristics. The Mann–Whitney U and Kruskal–Wallis H tests were used for the non-normal data. Finally, multi-level hierarchical regression analysis was performed at three levels. The first regression model (Model 1) was conducted for the demographic characteristic variables using each dependent variable to examine the explanatory power of demographic characteristic variables for job performance and the respective two factors. The second regression model (Model 2) included demographic characteristic variables and social network centrality indicators. After controlling for the effect of demographic variables, the increment of network characteristic values on the explanation of job performance and its two factors were investigated. The final regression model (Model 3) included demographic characteristic variables, social network centrality indicators, and CI. The increment of structural holes in the explanation of job performance and its respective two factors were examined by controlling for the effect of demographic characteristic variables and social network centrality indicators. Two-tailed *p* < 0.05 was considered significant.

## 3. Results

### 3.1. Sample Characteristics

There were 338 members in the hypertension management team in the three county medical alliances. Furthermore, 15.7% of the participants were employed in county-level health services institutions, 32.8% worked in township health services institutions, and 51.5% worked in village clinics. Participants with a maximum experience of 5 years accounted for 15.7%; 22.2% had 6–15 years of experience; 26.0% had 16–25 years of experience; and 36.1% had 26 years of experience. The general information distribution of the hypertension management team members in county medical alliances is shown in Table 1.

### 3.2. Eigenvalues of Social Network Characteristics

Among the four types of networks, the density of information network of the hypertension management team in Suixi County Hospital Medical Alliance was the highest (0.029), while the density of the friend network and trust network in Youxi County General Hospital Medical Alliance was the lowest (0.011). This means that the hypertension team was less connected with fewer interactions among members and less than 3% of all possible connections. The average distance of the trust network of Youxi County General Hospital Medical Alliance was the shortest (1.759), while the average distance of the advice network of Suixi County TCM Hospital Medical Alliance was the longest (5.852). This shows that the network between a member and any other member generally needs to be through one to six people to create a connection (Table 2 and Figure 1, Figure 2 and Figure 3).

### 3.3. Univariate Analysis

Table 3 shows the scores of each item of TP, CP, and overall job performance. It can be seen that the scores of each item of CP are mostly higher than that of TP.

Normal test analysis showed that TP, CP, and overall job performance had non-normal distributions (K–S test, TP: Z = 0.170, *p* < 0.001; CP: Z = 0.164, *p* < 0.001; overall job performance: Z = 0.134, *p* < 0.001). The median (interquartile spacing) describes all the factors. TP, CP, and job performance scored 4.25 (1.00), 4.60 (1.00), and 4.39 (0.89), respectively, which were generally at a high level. The self-rated CP of hypertension management team members was higher than TP. The results showed that the difference in TP according to years of service (H = 7.832, *p* = 0.050) was significant. Furthermore, the difference in CP according to the level of health services institutions served (H = 9.374, *p* = 0.009) was significant. Similarly, the difference in overall job performance according to the level of health services institutions served (H = 7.202, *p* = 0.027) was significant (Table 4).

### 3.4. Multiple Hierarchical Regression Analysis

Using TP, CP, and overall job performance as dependent variables, the results showed that R^2^ was significant (R^2^ = 0.148, F = 3.258, *p* < 0.001; R^2^ = 0.193, F = 4.503, *p* < 0.001; R^2^ = 0.184, F = 4.237, *p* < 0.001, respectively). The results showed that the level of health services institutions served, the in-closeness centrality of the advice network (β = 0.176, *p* < 0.05), and the betweenness centrality of the trust network (β = 0.126, *p* < 0.05) were positively associated with TP. CP was associated with the level of health services institutions served, the in-closeness centrality of the advice network (β = 0.226, *p* < 0.05), the out-closeness centrality of the information network (β = −0.368, *p* < 0.01), and the CI of the friendship network (β = 0.130, *p* < 0.05). Furthermore, a regression relationship of statistical significance was found among the level of health services institutions served, the in-closeness centrality of the advice network (β = 0.213, *p* < 0.05), the out-closeness centrality of the information network (β = −0.334, *p* < 0.05), the CI of the friendship network (β = 0.132, *p* < 0.05), and the overall job performance (Table 5).

## 4. Discussion

Based on an integrated perspective, this study explores the impact of the social network of NCD management teams formed by cross-organizational and multidisciplinary cooperation on the job performance of medical staff. The results suggest that ego network centrality and the abundance of structural holes affected the TP, CP, and overall job performance in the social networks of the hypertension management teams in county medical alliances. Specific findings are elaborated below.

### 4.1. Analysis of Job Performance

The hypertension management team members had a high job performance level of 4.39 (0.89), scoring higher than the general staff [34]. Owing to the unique nature of the medical profession, a higher standard of work skills is expected among the medical staff, while from an educational perspective, they are expected to have a sense of mission and responsibility. The CP score (4.60 (1.00)) was higher than the TP score (4.25 (1.00)). The self-assessed performance scores showed that most of the members expressed their work capability and expertise to be limited, despite their enthusiasm and willingness to additionally contribute at work. Thus, this study suggests that the inadequate capability of diagnosis and treatment in rural areas is an important factor limiting the level of hypertension management in county medical alliances [35]. Therefore, the cultivation of talents, enhancement of training efforts, and improvement in the quality of training should be prioritized in these alliances.

The CP and job performance of the team members in county-level hospitals were lower than those of the team members in the township health services institutions. County hospitals deal with a far greater number of patients than township health services institutions, leading to increased pressure, work stress, and occupational risks that influence staff’s work attitudes and patient-reception behaviors [36]. As a result, county hospital staff members are prone to increased tension and anxiety, which affect their service quality and job performance. In contrast to the team members in village clinics, the TP, CP, overall job performance, working capacity, and professional knowledge among the team members in township health services institutions were higher. The limitations in practical knowledge and skills, as well as working conditions that do not meet work needs, prevent staff in village clinics from accomplishing their job performance targets to the maximum [37]. Hence, it is crucial to establish corresponding compensation measures, raise salary levels, and provide reasonable insurance to encourage village doctors to provide high-quality chronic disease management services [38].

### 4.2. Differences in the Density of the Four Whole Networks

Among the four networks, the densities of the advice network and information network were relatively higher compared to the friendship network and trust network. This result indicated that there was a reduced exchange of feelings among team members. However, communication increased, and more work-related knowledge and information were disseminated among frequent work-related contacts. The highest density of the advice network was 0.027, and all possible contact existing among team members was less than 3%. This finding indicated that there was a lack of communication among the hypertension management team members in the three-county medical alliances. Knowledge exchange on hypertension management was fragmented, and the members worked relatively independently. This results in a lack of detailed labor division and the ineffective implementation of long-term collaboration mechanisms [39]. Most medical staff members under the current model of integrated NCD management are unable or unwilling to participate in cross-organization and multidisciplinary collaboration [40]. Thus, it is urgent to develop a coordination mechanism to regulate the cross-organization and multidisciplinary collaborative behavior among these medical staff members.

The information network showed a frequency that was slightly higher than that of the advice network but an average distance that was slightly lower than that of the advice network. This indicates that team members focus more on the transfer of simple work-related information during the process of multidisciplinary communication and interaction. Furthermore, there is insufficient communication and interaction surrounding complex forms of knowledge such as seeking work advice and solving work-related problems. As also observed in previous studies, the transfer of simple work-related information is easier than transferring technical work problems [41]. Therefore, robust guidance is required to develop a cross-organization and multidisciplinary advice network. This type of network can result in frequent and extensive communication, as well as cooperation between institutions and disciplines, to improve the management quality of hypertension management teams in China’s county medical alliances.

### 4.3. Social Network Characteristics and Job Performance

#### 4.3.1. TP Factor

The in-closeness centrality of the advice network and the betweenness centrality of the trust network varied in the same direction as TP. Higher levels of in-closeness centrality among the advice network members facilitate other team members to get to the nodes. This indicates the effectiveness and central position of the members who play a key role in the information distribution among other nodes in the network [42]. Thus, members can receive more knowledge about hypertension management, improve their problem-solving skills, control more resources, and demonstrate a robust integrative ability to address various problems related to hypertension management. Consistent with the current study, several previous studies have demonstrated that the centrality of the network has a strong correlation with and is an important predictor of performance [43,44].

Owing to the higher levels of betweenness centrality in the trust network, more members were required to act as a bridge in this network, which has a strong ability to control resources and occupy more key positions in the flow of resources and information. Furthermore, this results in a higher level of psychological security, an improved sense of well-being in hypertension management work, and a willingness to devote more time to improving work capability and job performance. Several studies have underscored the positive correlation between the bridge position of researchers in a network and their scientific performance [11]. A similar conclusion was derived in this study. A higher level of CI in the friendship network indicates an increase in team members in the node of the “bridge” that occupies information flow and psychological support, thus increasing the mobilization of social capital and heterogeneous information. Furthermore, the friendship network strives for more cooperation opportunities and the benefits of information and resource control owing to their advantageous network structural positions, subsequently improving their TP [45].

#### 4.3.2. CP Factor

The in-closeness centrality of the advice network and the CI of the friendship network varied in the same direction as CP, and the out-closeness of the information network varied inversely with CP. Team members with higher in-closeness centrality of their advice network were more open to member counseling. Increased professional knowledge and communication with other members can increase psychological support [46] and promote authority and status [16]. A previous study consistent with the results of the current research suggested that members among the advice network with high in-closeness centrality demonstrate an increased willingness to contribute to their work and improve friendly and reciprocal helping relations within the team, thus achieving higher CP [47,48]. Consequently, the CI of a friendship network refers to members at the “bridge” of this heterogeneous network. The position of the structural hole provides structural and intermediate advantages for the members. Increased psychological support and information conducive to work can be obtained through emotional exchange among friends, while simultaneously providing psychological support for other members, producing a greater external influence and improving team performance [49]. A high out-closeness centrality among the information network indicates that the team members are in the position of information providers, thus facilitating the reach of other nodes. Furthermore, they require more time to learn and understand information related to hypertension training and provide information to other members. However, exacerbated intelligence anxiety can create learning difficulties for the members [50], which may cause a gradual decline in enthusiasm, dedication, and communication with members in the work context (except for necessary information transfer), subsequently decreasing CP.

#### 4.3.3. Overall Job Performance

The in-closeness centrality of the advice network and the CI of the friendship network showed a homogeneous change with job performance, and the out-closeness of the information network changed inversely with job performance. Members with higher in-closeness centrality among the advice network receive more knowledge about hypertension management, possess more clinical knowledge and skills, and have better job performance. Similar to a study by Contandriopoulos et al. [11], the present study argues that the structural position in a collaborative network is positively associated with job performance. Members with a high CI in the friendship network are in the key position of informal communication, have high authority and influence in the work team, and can obtain external help and psychological support when they encounter difficulties, which is conducive to the improvement of their work performance. Members with high out-closeness centrality of information need to increase their knowledge of hypertension and provision to other members. This may cause the individual to invest more energy in communicating information, which may affect the completion of their job tasks.

### 4.4. Limitations

Owing to constraints in time and relevant resources, this study had several potential limitations. The first concerns the validity of the performance measure. Self-report measures of job performance are prone to various biases and are viewed skeptically in industrial–organizational psychology and allied fields [51]. However, because the hypertension management service is an integrated service, we could not find a way to evaluate the performance of this integration service on an individual basis. Furthermore, this study is consistent with the classic approach of social network research, which focuses on describing a network’s structural characteristics at specific time nodes without assessing temporal variation. Future studies should focus on the dynamic changes in cross-organization and multidisciplinary social networks among medical staff over time, as well as the impact of such changes on their cooperation.

## 5. Conclusions

Our study found that the social network of the NCD management team, especially the hypertension management team, has an impact on their members’ job performance. Communication and exchange among members are essential for improving individual job performance. Therefore, it is crucial to build an information- and resource-sharing mechanism, enhance the contact density among the network members to achieve more extensive connections, and foster in-depth communication to circulate professional knowledge among team members. Moreover, given the negative relationship between the out-closeness centrality of the information network and CP and overall job performance, in the process of NCD management cooperation, medical and health institutions should pay attention to setting up a special position for transferring information related to NCDs.

## Figures and Tables

**Figure 1 healthcare-11-02218-f001:**
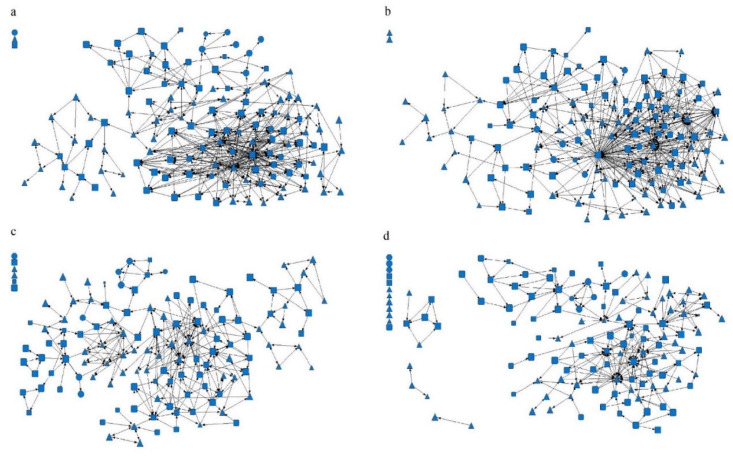
Suixi County Hospital Medical Alliance Social Network Relationship. Note: (**a**) Advice network, (**b**) information network, (**c**) friendship network, and (**d**) trust network. The circle represents the hypertension management medical staff of the county medical community, the square represents the hypertension management medical staff of the community health service center or health center, and the triangle represents the hypertension management medical staff of the village level. The graph size represents the job performance level of the medical staff.

**Figure 2 healthcare-11-02218-f002:**
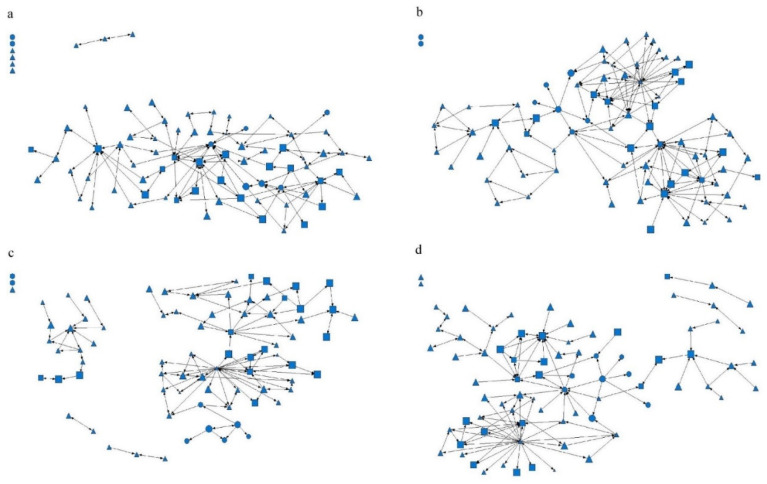
Suixi County TCM Hospital Medical Alliance Social Network Relationship. Note: (**a**) Advice network, (**b**) information network, (**c**) friendship network, and (**d**) trust network. The circle represents the hypertension management medical staff of the county medical community, the square represents the hypertension management medical staff of the community health service center or health center, and the triangle represents the hypertension management medical staff of the village level. The graph size represents the job performance level of the medical staff.

**Figure 3 healthcare-11-02218-f003:**
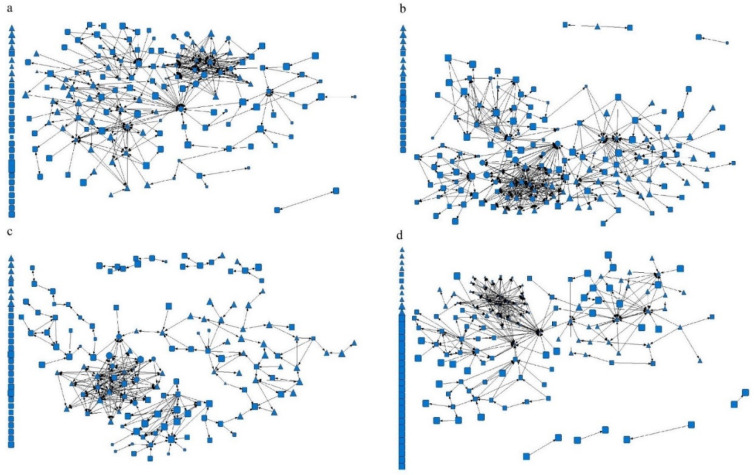
Youxi County General Hospital Medical Alliance Social Network Relationship. Note: (**a**) Advice network, (**b**) information network, (**c**) friendship network, and (**d**) trust network. The circle represents the hypertension management medical staff of the county medical community, the square represents the hypertension management medical staff of the community health service center or health center, and the triangle represents the hypertension management medical staff of the village level. The graph size represents the job performance level of the medical staff.

**Table 1 healthcare-11-02218-t001:** Descriptive Statistics.

Variable		*n*	%	Variable		*n*	%
Age	29 and below	49	14.5	Level of Health Services Institutions Served	County-level Hospitals	53	15.7
	30~39 years old	55	16.3		Township Health Services Institutions	111	32.8
	40~49 years	127	37.6		Village Clinics	174	51.5
	50 and above	107	31.6	Years of Service	5 years and below	53	15.7
Marital Status	Married	299	88.5		6~15 years	75	22.2
	Others (unmarried, divorced, widowed, and so on)	39	11.5		16~25 years	88	26.0
Health Conditions	Healthy	275	81.4		26 years and above	122	36.1
	Others (sub-health, illness, and so on)	63	18.6				

**Table 2 healthcare-11-02218-t002:** Whole Eigenvalue of Social Network.

County Medical Alliance	Type of Network	Density	Avg Distance
Suixi County Hospital Medical Alliance	Advice Network	0.027	5.291
	Information Network	0.029	3.846
	Friendship Network	0.017	5.587
	Trust Network	0.014	3.026
Suixi County TCM Hospital Medical Alliance	Advice Network	0.024	5.852
	Information Network	0.027	5.233
	Friendship Network	0.020	3.332
	Trust Network	0.019	4.319
Youxi County General Hospital Medical Alliance	Advice Network	0.013	3.942
	Information Network	0.016	3.653
	Friendship Network	0.011	2.606
	Trust Network	0.011	1.759

**Table 3 healthcare-11-02218-t003:** The score for each item of job performance.

Variable	Median	IQR
TP1	4.00	1.00
TP2	4.00	1.00
TP3	4.00	1.00
TP4	4.00	1.00
CP1	5.00	1.00
CP2	4.00	1.00
CP3	5.00	1.00
CP4	5.00	1.00
CP5	4.00	1.00
Overall job performance	4.39	0.89

Note: IQR: interquartile range.

**Table 4 healthcare-11-02218-t004:** Univariate Analysis.

Variable	Classification	TP Factor	CP Factor	Overall Job Performance
Age	29 and below	4.000 (1.250)	4.600 (0.900)	4.111 (1.000)
	30~39 years old	4.000 (1.000)	4.400 (0.800)	4.111 (0.670)
	40~49 years	4.250 (1.000)	4.600 (1.000)	4.444 (1.000)
	50 and above	4.250 (1.000)	4.600 (1.000)	4.333 (0.780)
H (*p*)		5.904 (0.116)	3.211 (0.360)	5.137 (0.162)
Level of Health Services Institution Served	County-level Hospitals	4.250 (1.500)	4.600 (1.200)	4.333 (1.220)
	Township Health Institutions	4.250 (1.000)	4.800 (0.800)	4.556 (1.000)
	Village Clinics	4.000 (1.000)	4.400 (0.800)	4.222 (0.780)
H (*p*)		4.673 (0.097)	9.374 (0.009)	7.202 (0.027)
Marital Status	Married	4.250 (1.000)	4.600 (1.000)	4.333 (0.890)
	Others	4.000 (0.750)	4.600 (0.800)	4.444 (0.780)
Z (*p*)		−0.686 (0.493)	−0.559 (0.576)	−0.721 (0.471)
Health Status	Healthy	4.250 (1.000)	4.600 (1.000)	4.333 (0.890)
	Others	4.250 (1.000)	4.600 (1.000)	4.444 (0.780)
Z (*p*)		−0.274 (0.784)	−1.002 (0.316)	−0.778 (0.436)
Years of Service	5 years and below	4.000 (1.000)	4.600 (0.800)	4.222 (0.890)
	6~15 years	4.000 (1.000)	4.400 (1.000)	4.111 (0.780)
	16~25 years	4.250 (1.000)	4.600 (0.800)	4.444 (0.780)
	26 years and above	4.125 (1.000)	4.600 (1.000)	4.444 (1.000)
H (*p*)		7.832 (0.050)	5.1119 (0.163)	7.323 (0.062)

Note: The *p*-value was set to two-tailed.

**Table 5 healthcare-11-02218-t005:** Multiple Linear Regression Analysis.

Variable	TP Factor			CP Factor			Overall Job Performance		
	Model 1	Model 2	Model 3	Model 1	Model 2	Model 3	Model 1	Model 2	Model 3
	β	β	β	β	β	β	β	β	β
Constant									
Level of Health Services Institutions served									
Village clinics (ref)									
County-level Hospitals	0.056	0.082	0.057	0.062	0.098	0.063	0.062	0.094	0.063
Township Health Institutions	0.192 **	0.274 **	0.238 **	0.228 ***	0.330 ***	0.276 ***	0.221 ***	0.318 ***	0.271 ***
In-closeness Centrality of Advice Network		0.156	0.176 *		0.187 *	0.226 **		0.181 *	0.213 *
Out-closeness Centrality of Information Network		−0.306 *	−0.263		−0.425 **	−0.368 **		−0.387 **	−0.334 *
Betweenness Centrality of Trust Network		0.142 *	0.126 *		0.108	0.091		0.130 *	0.112
CI of Friendship Network			0.121			0.130 *			0.132 *
R^2^	0.058	0.120	0.148	0.067	0.155	0.193	0.068	0.15	0.184
Adjusted R^2^	0.058	0.062	0.028	0.067	0.089	0.038	0.068	0.082	0.034
F	5.096	3.387	3.258	5.936	4.585	4.503	6.071	4.385	4.237
Adjusted F	5.096	2.533	2.618	5.936	3.786	3.733	6.071	3.457	3.341
*p*-value of the model	0.001	<0.001	<0.001	<0.001	<0.001	<0.001	<0.001	<0.001	<0.001
VIFmax	1.601	6.594	6.901	1.601	6.594	6.901	1.601	6.594	6.901

Ref: reference group. * *p* < 0.05, ** *p* < 0.01, and *** *p* < 0.001.

## Data Availability

The datasets used and/or analyzed during the current study are available from the corresponding author on reasonable request.

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
