# Peer review of "Effects of Social Networks on Job Performance of Individuals among the Hypertension Management Teams in Rural China"

_healthcare, 2023, doi:10.3390/healthcare11152218_

Round 1

Reviewer 1 Report

Thank you for inviting me to review this manuscript entitled "Effects of social networks on job performance of individuals: A study of hypertension management teams in rural China". This study conducted an in-depth examination and validation of the influence of complex cross-organization and multidisciplinary social networks on the job performance of team members.

Some comments were raised and must be addressed 

Abstract: according to the guideline, the abstract must be structured. So, please reconstruct the abstract showing background, aim, method, results, and conclusion.

The results of the abstract must be presented in numbers. 

Introduction:

"For the cooperation of medical institutions, corresponding functional teams are set up to better promote goal realization, and a network of cross-organizational and multi-disciplinary communication and cooperation is gradually formed." this sentence is not clear enough, please re-write it.

in the introduction, the author did not discuss "hypertension" and the research gap behind it? please elaborate more. 

methods:

how did the authors calculate the sample size? 

did the authors used the published questionnaire as it is?

what was the administration language?

if they performed any changes, what was it?

did they perform any kind of piloting or validation?

what about the consent form?

please write more details about data collection and recruitment.

results:

great presentation for the results

Discssuion

please elaborate in this section "Members with a high CI in the friendship network demonstrate increased information sharing and psychological support, authority and influence in the team, and access to external help when they encounter difficulties, thus improving their job performance"

conclusion:

please one master sentence, that answers the research question and aim before the implications

Author Response

1. Abstract: according to the guideline, the abstract must be structured. So, please reconstruct the abstract showing background, aim, method, results, and conclusion.

The results of the abstract must be presented in numbers.

Re: Thank you for your constructive comments. We have reconstructed the abstract showing background, aim, method, results, and conclusion. The results of the abstract have been presented in numbers (Page 1, lines 9-25). Background Limited studies have explored the relationship among cross-organizational and multidisciplinary medical staff. Aim The present study conducts an in-depth examination and validation of the influence of complex cross-organization and multidisciplinary social networks on the job performance of team members. Method Multi-level hierarchical regression analysis was used to assess the impact of the centrality and the characteristics of structural holes in social networks (i.e., advice network, information network, friendship network, and trust network) on job performance. Results The in-closeness centrality of the advice network (β=0.176, p<0.05) and the betweenness centrality of the trust network (β=0.126, p<0.05) have positive effects on task performance. The in-closeness centrality of the advice network (β=0.226, p<0.05; β=0.213, p<0.05) and the CI (1– constraint index) of the friendship network (β=0.130, p<0.05; β=0.132, p<0.05) have positive effects on contextual performance and overall job performance. Meanwhile, the out-closeness centrality of the information network (β= – 0.368, p<0.01; β= – 0.334, p<0.05) has a negative effect on contextual performance and overall job performance. Conclusion This study investigates the relationship between healthcare professionals' job performance and their social networks, taking into account the perspectives of cross-organizational and multidisciplinary teams. The study contributes to the effort of breaking down barriers between different disciplines and organizations, and ultimately, improving the quality of healthcare delivery.

2. Introduction:

"For the cooperation of medical institutions, corresponding functional teams are set up to better promote goal realization, and a network of cross-organizational and multi-disciplinary communication and cooperation is gradually formed." this sentence is not clear enough, please re-write it.

Re: Thank you for your suggestion. We have re-written this sentence to make it easier to understand (Page 1, lines 40-43). In order to better promote the realization of the goal, medical institutions establish corresponding functional teams through mutual cooperation, and gradually form a cross-organizational and multidisciplinary communication and cooperation network in the process.

in the introduction, the author did not discuss "hypertension" and the research gap behind it? please elaborate more.

Re: Thank you for your constructive comments and suggestions. We have added more information about "hypertension"(Page 2, lines 72-85). In the medical field, structural integration and functional integration have been extensively studied, while research on process and interpersonal integration needs to be deepened [1]. At the same time, the research on integration effect is scarce. Studying the existing problems of cross-organizational and multidisciplinary cooperation is essential to improve the performance levels. In addition, hypertension is a major public health concern globally with an increasingly rapid prevalence in low- and middle-income countries [2]. At present, hypertension management in rural China needs to be addressed through cross-organizational and multidisciplinary collaboration. However, few studies have explored the current situation and effectiveness of communication and cooperation between NCDs management teams, especially hypertension management teams. Therefore, starting from NCDs, this study uses complex network theory to explore the impact of cross-organizational and multidisciplinary cooperation of social networks on the job performance of medical staff, to provide evidence support for the integration effect and reference for future research.

3.methods:

how did the authors calculate the sample size?

Re: Thank you for your questions. There are views and studies that the sample size should be more than 300, or even about 150 is enough [3,4]. The participants of this study were reported by the personnel management department of the county medical alliance, and reached the requirement of more than 300.

did the authors used the published questionnaire as it is?

Re: Thank you for your questions. In this survey, we used the published questionnaire as it is. The information on the job performance of the hypertension management team members was collected using the Job Performance Scale based on the scale designed by Han [5]. In the social network part of the study, Krackhardt and Jiade Luo's classification method and item setting were used.

what was the administration language?

Re: Thank you for your questions. The administration language was Chinese.

if they performed any changes, what was it?

Re: Thank you for your questions. We regret to inform you that because this study is a cross-sectional study, it is not possible to report changes in job performance of members of the hypertension management team.

did they perform any kind of piloting or validation?

Re: Thank you for your questions. The study was a cross-sectional study and was not piloting or validation. However, the questionnaire was developed with reference to previous maturity scales. In addition, to test the reliability and validity of the scale, we conducted confirmatory factor analysis and model fit test (Page 15-17).

what about the consent form?

Re: Thank you for your questions. We have sent the consent form and ethics review to the editor.

please write more details about data collection and recruitment.

Re: Thank you for your suggestion. We have write more details about data collection and recruitment (Page 3, lines 113-123). The data in this study were obtained through questionnaires conducted between April 2021 and June 2021. The research team requested the responsible unit of the local county medical alliances to provide a specific list of people involved in hypertension management, and to communicate in advance to ensure that the study subjects were aware of the research project and actively participated in the investigation. The lists of members of the hypertension management teams provided by the personnel departments of the county medical alliances were verified, added or removed, and coded to ensure that the SNA was conducted on a complete group. Anonymity and confidentiality were emphasized during the data collection process, and the questionnaires were received on-site. A total of 382 questionnaires were distributed, of which 338 were completed and returned, resulting in a response rate of 88.48%.

4. results:

great presentation for the results

Re: Thank you for your positive comments.

5. Discussion

please elaborate in this section "Members with a high CI in the friendship network demonstrate increased information sharing and psychological support, authority and influence in the team, and access to external help when they encounter difficulties, thus improving their job performance"

Re: Thank you for your suggestion. We have elaborated in this section (Page 11, lines 382-385). Members with a high CI in the friendship network are in the key position of informal communication, have high authority and influence in the work team, and can obtain external help and psychological support when they encounter difficulties, which is conducive to the improvement of their work performance.

6. conclusion:

please one master sentence, that answers the research question and aim before the implications

Re: Thank you for your suggestion. We have modified this section accordingly (Page 12, lines 402-411). Our study found that the social network of NCDs management team, especially hypertension management team, has an impact on their members' job performance. Communication and exchange among members are essential for improving individual job performance. Therefore, it is crucial to build an information and resource sharing mechanism, enhance the contact density among the network members to achieve more extensive connections, and foster in-depth communication to circulate professional knowledge among team members. Moreover, given the negative relationship between the out-closeness centrality of the information network and CP and overall job performance, in the process of NCDs management cooperation, medical and health institutions should pay attention to setting up a special position for transferring information related to NCDs.

Reference

1.Burns LR, Asch D, Muller R. Vertical integration of physicians and hospitals: Three decades of futility? Cambridge U.K.: Cambridge University Press; 2022.

2.Geldsetzer P, Manne-Goehler J, Marcus ME, Ebert C, Zhumadilov Z, Wesseh CS, Tsabedze L, Supiyev A, Sturua L, Bahen-deka SK, et al. The state of hypertension care in 44 low-income and middle-income countries: A cross-sectional study of nationally representative individual-level data from 11 million adults. Lancet. 2019, 394, 652-662.

3.Stevens J. Applied multivariate statistics for the social sciences. Journal of educational statistics, 2002, 47(2).

4.Barbara G. Tabachnick, Linda. S. Fidell. Using multivariate statistics (5th ed.). Pearson/Allyn & Bacon, 2007.

5.Han Y, Liao J. Task performance and contextual performance based on performance separability. Industrial Engineering Journal. 2006, 49-53.

Reviewer 2 Report

This study uses complex network theory to explore the impact of cross-organizational and multidisciplinary cooperation of social networks on the job performance of medical staff, to provide evidence support for the integration effect and reference for future research. Moreover, the study contributes to the effort of breaking down barriers between different disciplines and organizations, and ultimately, improving the quality of healthcare delivery.

1)    This study is novel and interesting and also covers a gap in the current literature.

2)    The Introduction is well written and adequate reference to other relevant papers has been made.

3)    Methods are robust and valid.

4)    Results are very well detailed and are very relevant and impressive.

5)    Discussion is well elaborated with a good insight and critical point of view.

Overall considered, the authors should be congratulated for this article.

Author Response

This study uses complex network theory to explore the impact of cross-organizational and multidisciplinary cooperation of social networks on the job performance of medical staff, to provide evidence support for the integration effect and reference for future research. Moreover, the study contributes to the effort of breaking down barriers between different disciplines and organizations, and ultimately, improving the quality of healthcare delivery.

1)    This study is novel and interesting and also covers a gap in the current literature.

2)    The Introduction is well written and adequate reference to other relevant papers has been made.

3)    Methods are robust and valid.

4)    Results are very well detailed and are very relevant and impressive.

5)    Discussion is well elaborated with a good insight and critical point of view.

Overall considered, the authors should be congratulated for this article.

Re: Thank you for your positive comments.

Reviewer 3 Report

Dear authors,

I find the focus of your study relevant. However, I suggest that you improve on the following areas:

1. Title- the title of the study is not straightforward. The way it is phrased appear disconnected. 

Effects of social networks on job performance of individuals: A study of hypertension management teams in rural China

The first phrase  ( Effects of social networks on job performance of individuals: ) seems to be a different study from the second phrase ( A study of hypertension management teams in rural China.

I suggest for you to rephrase the title.

2. Abstract: The first part of the abstract should already be able to present the population and setting of the study. The presentation is so generic and does not lead the reader to what or where the study is directed. I suggest the the aim of the study be moved to the beginning of the abstract rather than at the end to set the focus of the study and therefore be able to relate the results presented.

3. Introduction - I suggest to reorganize the transition of how the concepts ( are  presented 

I found different concepts discussed in the introduction  ( Integrated health care, SCP, Social network) however, the main focus of the study was not well established. 

I suggest that the main focus of the study be well defined and established from the start. The introduction part did not give justification or connection to the results presented.

The aims of the study presented in the abstract (lines 19-23, introduction (Lines 77-80), and methodology (lines 100-103) (lines 140-142)  are not consistent.

4. Methods- The introduction to the research design should be better written in the introduction section.  I further suggest that the Methodolgy section should be reorganized with subheadings (Design, population/sample, setting, etc).

The research design was not clearly identified.

Instruments/Measures: 

    - Under the job performance variable- what are TPs and CPs? How is the qualitative description of  job performance? Only the likert scale was described, how is the score interpreted and what are cut off points for the job performance?

Statistical analysis- Divide the one long paragraph into at least 2 to 3 paragraphs separating the descriptive statistics from the inferential statistics.

Explain the meaning of density and average distance

Results -

   What are others in the demographic profiles (marital status and health conditions)?

There should be a qualitative description of the job performance of the participants.

Present in a separate table the scales and scores and its equivalent qualitative description for the TP and CP.

Conclusion- The conclusion regarding communication and exchange to enhance job performance were not reflected in the results.

The conclusion should address the aims of the study and be based from the results and analysis.

Over all, I find the organization of the paper quite hard to follow because the discussion and transition of the different sections (introduction, methodology, and results) are disconnected. Although the topic is relevant, the paper needs to be rewritten.

The thesis of the study should be well established and the variables be well-defined. A brief introduction or description of the statistics should be able to help the readers understand the results presented.

Thank you and good luck

I suggest an English language editing.

Author Response

1.Title- the title of the study is not straightforward. The way it is phrased appear disconnected.

Effects of social networks on job performance of individuals: A study of hypertension management teams in rural China

The first phrase (Effects of social networks on job performance of individuals: ) seems to be a different study from the second phrase ( A study of hypertension management teams in rural China.

I suggest for you to rephrase the title.

Re: Thank you for your suggestion. We have rephrased the title to " Effects of social networks on job performance of individuals: A study based on hypertension management teams in rural China".

2.Abstract: The first part of the abstract should already be able to present the population and setting of the study. The presentation is so generic and does not lead the reader to what or where the study is directed. I suggest the aim of the study be moved to the beginning of the abstract rather than at the end to set the focus of the study and therefore be able to relate the results presented.

Re: Thank you for your suggestion. I have structured the abstract section as suggested, highlighting the aim of the study (Page 1, lines 9-25). Background Limited studies have explored the relationship among cross-organizational and multidisciplinary medical staff. Aim The present study conducts an in-depth examination and validation of the influence of complex cross-organization and multidisciplinary social networks on the job performance of team members. Method Multi-level hierarchical regression analysis was used to assess the impact of the centrality and the characteristics of structural holes in social networks (i.e., advice network, information network, friendship network, and trust network) on job performance. Results The in-closeness centrality of the advice network (β=0.176, p<0.05) and the betweenness centrality of the trust network (β=0.126, p<0.05) have positive effects on task performance. The in-closeness centrality of the advice network (β=0.226, p<0.05; β=0.213, p<0.05) and the CI (1- constraint index) of the friendship network (β=0.130, p<0.05; β=0.132, p<0.05) have positive effects on contextual performance and overall job performance. Meanwhile, the out-closeness centrality of the information network (β= –0.368, p<0.01; β= –0.334, p<0.05) has a negative effect on contextual performance and overall job performance. Conclusion This study investigates the relationship between healthcare professionals' job performance and their social networks, taking into account the perspectives of cross-organizational and multidisciplinary teams. The study contributes to the effort of breaking down barriers between different disciplines and organizations, and ultimately, improving the quality of healthcare delivery.

3.Introduction - I suggest to reorganize the transition of how the concepts ( are presented

I found different concepts discussed in the introduction (Integrated health care, SCP, Social network) however, the main focus of the study was not well established.

I suggest that the main focus of the study be well defined and established from the start. The introduction part did not give justification or connection to the results presented.

Re: Thank you for your suggestion. We have adjusted the logical order of the preface accordingly. We have revised the presentation in the introduction to make the logic clearer. The concept in the introduction of this study is introduced by integrated healthcare, which is the current situation of cooperation among medical institutions. In the process of cooperation, various forms of cooperation appear, and cross-organizational and multidisciplinary cooperation is an important content. In the process of cooperation, the social network of team members is gradually formed and may have an impact on their job performance (Based on the Structure-Conduct-Performance analysis paradigm). Therefore, we use social network analysis method to explore the impact of social network characteristics on members' job performance.

The aims of the study presented in the abstract (lines 19-23, introduction (Lines 77-80), and methodology (lines 100-103) (lines 140-142) are not consistent.

Re: Thank you for your suggestion. The aim of the study is to explore the effects of the types and structural characteristics of cross-organizational and multidisciplinary cooperation social networks on the job performance of medical staff.

4.Methods- The introduction to the research design should be better written in the introduction section. I further suggest that the Methodolgy section should be reorganized with subheadings (Design, population/sample, setting, etc).

The research design was not clearly identified.

Re: Thank you for your suggestion. We have revised the research design section and reorganized it by subheadings (Page 2, lines 88-194).

5.Instruments/Measures:

- Under the job performance variable- what are TPs and CPs? How is the qualitative description of job performance? Only the likert scale was described, how is the score interpreted and what are cut off points for the job performance?

Re: Thank you for your questions. We have added the concepts of TP and CP to "2.3.1 job performance" section and changed the relevant expressions (Page 3, lines 127-138). Job performance is an important factor affecting the effectiveness of hypertension prevention and control. Job performance refers to the sum of behaviors and results related to organizational goals within a certain period. TP refers to employee behaviors that are directly related to the organization and technology, reflecting the maintenance and service work that realizes technical needs. CP refers to voluntary employee behaviors that can facilitate the organization and its members to achieve organizational goals. In this study, the job performance of hyper-tension management team members was measured with two factors–TP (four entries) and CP (six entries). Overall job performance is the sum of TP and CP scores. Each option was scored using a five-point Likert scale (1 = “strongly disagree,” 2 = “disagree,” 3 = “neither agree nor disagree,” 4 = “agree,” or 5= “strongly agree”). The score is calculated by the mean of each item. The higher the score, the higher the job performance.

Statistical analysis Divide the one long paragraph into at least 2 to 3 paragraphs separating the descriptive statistics from the inferential statistics.

Re: Thank you for your questions. We have divided the one long paragraph into 2 paragraphs.

Explain the meaning of density and average distance

Re: Thank you for your suggestion. The meaning and calculation method of density, average distance, degree centrality, betweenness centrality, closeness centrality and structural holes indicator were presented in Additional file 1 (Page  18).

6.Results -

What are others in the demographic profiles (marital status and health conditions)?

Re: Thank you for your questions. Others in the marital status means unmarried, divorced, widowed and so on. Others in the health conditions means sub-health, illness and so on.

There should be a qualitative description of the job performance of the participants.

Re: Thank you for your suggestion. We have been a qualitative description of the job performance of the participants (Page 7, lines 235-244). Normal test analysis showed that TP, CP, and overall job performance had non-normal distributions (K-S test, TP: Z = 0.170, p < 0.001; CP: Z = 0.164, p < 0.001; overall job performance: Z = 0.134, p < 0.001). The median (interquartile spacing) describes all the factors. TP, CP, and job performance scored 4.25 (1.00), 4.60 (1.00), and 4.39 (0.89), respectively, which were generally at a high level. The self-rated CP of hypertension management team members was higher than TP. The results showed that the difference in TP according to years of service (H = 7.832, p = 0.050) was significant. Furthermore, the difference in CP according to the level of health services institutions served (H = 9.374, p = 0.009) was significant. Similarly, the difference in overall job performance according to the level of health services institutions served (H = 7.202, p = 0.027) was significant.

Present in a separate table the scales and scores and its equivalent qualitative description for the TP and CP.

Re: Thank you for your suggestion. We have presented in a separate table the scales and scores and its equivalent qualitative description for the TP and CP in Supplementary Material (Page 16, 17).

7.Conclusion- The conclusion regarding communication and exchange to enhance job performance were not reflected in the results.

The conclusion should address the aims of the study and be based from the results and analysis.

Re: Thank you for your suggestion. Social network is a process of quantifying and evaluating the relationship between network members, that is, it measures the status quo of communication and cooperation among network members to a certain extent. We have revised the conclusion for better understanding (Page 12, lines 402-411). Our study found that the social network of NCDs management team, especially hypertension management team, has an impact on their members' job performance. Communication and exchange among members are essential for improving individual job performance. Therefore, it is crucial to build an information and resource sharing mechanism, enhance the contact density among the network members to achieve more extensive connections, and foster in-depth communication to circulate professional knowledge among team members. Moreover, given the negative relationship between the out-closeness centrality of the information network and CP and overall job performance, in the process of NCDs management cooperation, medical and health institutions should pay attention to setting up a special position for transferring information related to NCDs.

Round 2

Reviewer 3 Report

Dear Authors,

Thank you for your revised paper. In order to further improve your paper, I found the following areas that need to be improved more:

1. Title - May I suggest the title to be:

Effects of social networks on the  job performance of individuals among the  hypertension management teams in rural China

2. Abstract- Please be mindful of the verb tenses as this study was done already.

3. Introduction-The gap in the job performance and social network among the hypertension management team that pushed the study has not been clearly discussed.

4. Methods- The authors should remain faithful to the meaning of the subheadings (Design, population, etc)

Design- You discussed here the setting/locale and participants of the study but not the research design.

Population- You discussed here the data gathering procedure and ethical consideration.  Readers should should be able to understand who were your participants. Are they doctors? Nurses? How were they selected? What were your inclusion criteria?

Setting- This should refer to the locale where the study was conducted and not the variables of your study.

What are your references for Job performance and social network sections under methods?

The aim of the study stated in lines 82-86 with the statement in lines 156-159. Kindly reconcile.

Demographic profiles were not discussed in the introduction as part of the thesis.

5. Results-

Demographic profiles:

What do you mean by others in marital status and health conditions? State them in parentheses.

What are the professions of the members of the hypertension management team?

3.2 Eigenvalues- you only identified in your narratives the highest value for density but did not mention which among the four types of social network and from which county medical alliance. 

You should explain what density means or implies in terms of the social network relationship among the hypertension management team.  What is the interpretation of high and low densities in terms of the social network? The same with the average distance. How do you interpret the results?

You just presented the result of the statistical test but there is no interpretation or explanation for the readers to understand its meaning.

Although the discussion section made some clarifications, the interpretation should at least be explained as the results are presented.

3.3 Univariate analysis

What is your reference for qualifying values  for your TP, CP, and Job performance as high? What are your cut off points?

3.4- Kindly provide an interpretation for the  in-closeness, betweenness, and out-closeness centralities. What do they mean or imply on your specific variables.

Discussion:

The discussion including specific  items of TPs , CPs, and job performance create confusion since the specific items/scales/dimensions were not presented in the results.

I suggest to add a new table under the results section presenting the values/scores for each item of the TPs (4 items) and CPs (6 items), and the over all scores/values of the job performance. Only then can the discussion part be better appreciated.

Thank you and good luck.

There is still a need to edit the grammar in the paper.

Author Response

Dear Authors,

Thank you for your revised paper. In order to further improve your paper, I found the following areas that need to be improved more:

  1. Title - May I suggest the title to be:

Effects of social networks on the  job performance of individuals among the  hypertension management teams in rural China

Re: Thank you for your advice. We have modified the title to: Effects of social networks on the job performance of individuals among the hypertension management teams in rural China.

  1. Abstract- Please be mindful of the verb tenses as this study was done already.

Re: Thank you for your suggestion. We have modified the verb tenses will carefully. Background Limited studies have explored the relationship among cross-organizational and multidisciplinary medical staff. Aim The present study conducted an in-depth examination and validation of the influence of complex cross-organization and multidisciplinary social networks on the job performance of team members. Method Multi-level hierarchical regression analysis was used to assess the impact of the centrality and the characteristics of structural holes in social networks (i.e., advice network, information network, friendship network, and trust network) on job performance. Results The in-closeness centrality of the advice network (β=0.176, p<0.05) and the betweenness centrality of the trust network (β=0.126, p<0.05) had positive effects on task performance. The in-closeness centrality of the advice network (β=0.226, p<0.05; β=0.213, p<0.05) and the CI (1– constraint index) of the friendship network (β=0.130, p<0.05; β=0.132, p<0.05) had positive effects on contextual performance and overall job performance. Meanwhile, the out-closeness centrality of the information network (β= –0.368, p<0.01; β= –0.334, p<0.05) had a negative effect on contextual performance and overall job performance. Conclusion This study investigates the relationship between healthcare professionals' job performance and their social networks, taking into account the perspectives of cross-organizational and multidisciplinary teams. The study con-tributes to the effort of breaking down barriers between different disciplines and organizations, and ultimately, improving the quality of healthcare delivery.

  1. Introduction-The gap in the job performance and social network among the hypertension management team that pushed the study has not been clearly discussed.

Re: Thank you for your comments. We have added this to the introduction (Page 2, Lines 72-89). In the medical field, structural integration and functional integration have been extensively studied, while research on process and interpersonal integration needs to be deepened [1]. At the same time, the research on integration effect is scarce. Studying the existing problems of cross-organizational and multidisciplinary cooperation is essential to improve the performance levels. In addition, hypertension is a major public health concern globally with an increasingly rapid prevalence in low- and middle-income countries [2]. At present, hypertension management in rural China needs to be addressed through cross-organizational and multidisciplinary collaboration. However, few studies have explored the current situation and effectiveness of communication and cooperation between NCDs management teams, especially hypertension management teams. Previous studies have underscored the impact of social network characteristics on job performance, which provides feasibility for the present research. Additionally, Fu [3] demonstrated the theoretical scientific validity of studying the effect of social networks on the job performance of hypertension management teams at a county, township, and village level. Therefore, starting from NCDs, this study uses complex network theory to explore the impact of cross-organizational and multidisciplinary cooperation social networks on the job performance of medical staff, to provide evidence support for the integration effect and reference for future research.

  1. Methods- The authors should remain faithful to the meaning of the subheadings (Design, population, etc)

Design- You discussed here the setting/locale and participants of the study but not the research design.

Population- You discussed here the data gathering procedure and ethical consideration.  Readers should should be able to understand who were your participants. Are they doctors? Nurses? How were they selected? What were your inclusion criteria?

Setting- This should refer to the locale where the study was conducted and not the variables of your study.

Re: Thank you for your comments. We have restructured “Methods” to Study design and participants, Measures, and Statistical Analysis (Page 2-5, Lines 91-221).

What are your references for Job performance and social network sections under methods?

Re: Thank you. We have added references to each section (Page3, Lines 119-134).

2.2.1. Job Performance

Job performance refers to the sum of behaviors and results related to organizational goals within a certain period. TP refers to employee behaviors that are directly related to the organization and technology, reflecting the maintenance and service work that realizes technical needs. CP (contextual performance) refers to voluntary employee behaviors that can facilitate the organization and its members to achieve organizational goals. According to the job performance scale designed by Han [4], two factors–TP (four entries) and CP (five entries) was used in this study to measure the job performance of hypertension management team members. Overall job performance is the sum of TP and CP scores. Each option was scored using a five-point Likert scale (1 = “strongly disagree,” 2 = “disagree,” 3 = “neither agree nor disagree,” 4 = “agree,” or 5= “strongly agree”). The score is calculated by the mean of each item. The higher the score, the higher the job performance.

2.2.2. Social Network

The social networks in this study were classified into four types based on the social network classification methods proposed by Krackhardt and Jiade Luo: the advice network, information network, friendship network, and trust network [5].

The aim of the study stated in lines 82-86 with the statement in lines 156-159. Kindly reconcile.

Re: Thank you. After adjustment, duplicate parts have been removed.

Demographic profiles were not discussed in the introduction as part of the thesis.

Re: Thank you for your comments. Demographic data is only the control variable of our study, not the focus of the study, so it is only briefly described in the methods and discussion sections. We have adjusted this section to the end of 2.2. Measures (Page 3, Lines 115-118).

  1. Results-

Demographic profiles:

What do you mean by others in marital status and health conditions? State them in parentheses.

Re: Thank you for your comments. We've already stated the meaning by others in marital status and health conditions in parentheses.

What are the professions of the members of the hypertension management team?

Re: Thank you for your question. The number of physicians and nurses included in this study was uneven and not described in the results section. Medical practitioners accounted for 33.1% (112), assistant medical practitioners for 26.6% (90), nurse practitioners for 10.1% (34), and others for 30.2% (102).

3.2 Eigenvalues- you only identified in your narratives the highest value for density but did not mention which among the four types of social network and from which county medical alliance. 

You should explain what density means or implies in terms of the social network relationship among the hypertension management team.  What is the interpretation of high and low densities in terms of the social network? The same with the average distance. How do you interpret the results?

You just presented the result of the statistical test but there is no interpretation or explanation for the readers to understand its meaning.

Although the discussion section made some clarifications, the interpretation should at least be explained as the results are presented.

Re: Thank you for your comments. I have reinterpreted the results of this section (Page 6, Lines 233-242). Among the four types of networks, the density of information network of hyper-tension management team in Suixi County Hospital Medical Alliance was the highest (0.029), while the density of friend network and trust network in Youxi County General Hospital Medical Alliance was the lowest (0.011). This means that the hypertension team was less connected, with fewer interactions among members, less than 3% of all possible connections. The average distance of the trust network of Youxi County General Hospital Medical Alliance was the shortest (1.759), while the average distance of the advice network of Suixi County TCM hospital Medical Alliance was the longest (5.852). This shows that in the network between a member and any other member generally need to be through 1 to 6 people to create a connection.

3.3 Univariate analysis

What is your reference for qualifying values  for your TP, CP, and Job performance as high? What are your cut off points?

Re: Thank you for your question. Because job performance is scored on a scale of 1 to 5, with 4 being at a higher level. Therefore, we judged its score as high. Job performance scale is a self-rating scale, which is also a limitation of our research. We will pay attention to and correct this problem in the subsequent research.

3.4- Kindly provide an interpretation for the  in-closeness, betweenness, and out-closeness centralities. What do they mean or imply on your specific variables.

Re: Thank you for your advice. We added the concepts from this section to 2.2.2. Social Network.

Discussion:

The discussion including specific  items of TPs , CPs, and job performance create confusion since the specific items/scales/dimensions were not presented in the results.

I suggest to add a new table under the results section presenting the values/scores for each item of the TPs (4 items) and CPs (6 items), and the over all scores/values of the job performance. Only then can the discussion part be better appreciated.

Re: Thank you for your suggestions, in order to make the article easier to understand, we have made corresponding changes in the results section (Page 8, Lines 266-269). Table 1 shows the scores of each item of TP, CP and overall job performance. It can be seen that the scores of each item of CP are mostly higher than that of TP.

Table 1. Score for each item of job performance.

Variable

Median

IQR

TP1

4.00

1.00

TP2

4.00

1.00

TP3

4.00

1.00

TP4

4.00

1.00

CP1

5.00

1.00

CP2

4.00

1.00

CP3

5.00

1.00

CP4

5.00

1.00

CP5

4.00

1.00

Overall job performance

4.39

0.89

Note: IQR: interquartile range.

Reference:

  1. Burns LR, Asch D, Muller R. Vertical integration of physicians and hospitals: Three decades of futility? Cambridge U.K.: Cambridge University Press; 2022.
  2. Geldsetzer P, Manne-Goehler J, Marcus ME, Ebert C, Zhumadilov Z, Wesseh CS, Tsabedze L, Supiyev A, Sturua L, Bahen-deka SK, et al. The state of hypertension care in 44 low-income and middle-income countries: A cross-sectional study of nationally representative individual-level data from 1·1 million adults. Lancet. 2019, 394, 652-662.
  3. Fu H. Study of the impact of social network of rural basic public health service workers on job performance. Huazhong University of Science and Technology. 2018.
  4. Han Y, Liao J. Task performance and contextual performance based on performance separability. Industrial Engineering Journal. 2006, 49-53.
  5. Krackhardt, D. The strength of strong ties: The importance of philos in organizations. In N. Nohria, & Eccles, R. (Ed.), Networks and organizations: Structure, form, and action. Boston: Harvard Business School Press.1992. 

Round 3

Reviewer 3 Report

Dear Authors,

Thank you for the revised manuscript. It has been improved but I have just few minor comments:

Revised Title:

Effects of social networks on job performance of individuals on the hypertension management teams in rural China

Suggestion:

Effects of social networks on job performance of individuals among the hypertension management teams in rural China

Methods:

Study design and participants:

I suggest that you divide the current paragraph into two paragraphs. One paragraph for the study design and another paragraph for the participants instead of one long paragraph for both.

Thank you and good luck.

There are few minor grammatical errors. Kindly read carefully to address this aspect.

Author Response

  1. Revised Title:

Effects of social networks on job performance of individuals on the hypertension management teams in rural China

Suggestion:

Effects of social networks on job performance of individuals among the hypertension management teams in rural China

Re: Thank you for your suggestion. We have rephrased the title to " Effects of social networks on job performance of individuals among the hypertension management teams in rural China".

  1. Methods:

Study design and participants:

I suggest that you divide the current paragraph into two paragraphs. One paragraph for the study design and another paragraph for the participants instead of one long paragraph for both.

Re: Thank you for your suggestion. I have divided the current paragraph into two paragraphs (Page 2-3, lines 93-114).

A cross-sectional survey was conducted from April 2021 and June 2021. The county medical alliance developed the integrated three-level county territory medical service system. This system underscored the exploration of the integrated county-township-village management of a county territory with “county-level hospitals as leaders, township health centers as hubs, and village health clinics as bases,” as well as effective connection with the township-village integration. In this study, the World Bank China Health Reform Program-for-Results provinces -- Fujian and Anhui, were randomly selected as a county medical alliance pilot county (Suixi County in Anhui Province and Youxi County in Fujian Province). A quantitative analysis was conducted on the social networks of the hypertension management teams in Suixi County Hospital Medical Alliance, Suixi County Traditional Chinese Medicine Hospital Medical Alliance, and Youxi County General Hospital Medical Alliance. Ultimately, we conducted a questionnaire survey based on the hypertension management team member list provided by the personnel department of the county medical alliance.

Members of the hypertension management team of the county medical alliance, such as physicians from cardiology departments of county-level hospitals, general practitioners, nurses and public health physicians from township hospitals or community health service centers, and village doctors from village clinics, were selected as the study participants. Further, we completed drawing and numbering all the codes of the study participants in the investigation. A total of 382 questionnaires were distributed, of which 338 were completed and returned, resulting in a response rate of 88.48%. Ethical approval for this study was obtained from the Ethics Committee of Hangzhou Normal University.